# Neurophysiological Effects of Electrical Stimulation on a Patient with Neurogenic Bowel Dysfunction and Cauda Equina Syndrome after Spinal Anesthesia: A Case Report

**DOI:** 10.3390/medicina59030588

**Published:** 2023-03-16

**Authors:** Seung-Kyu Lim, Chang Han Lee, Min-Kyun Oh, Se-Woong Chun

**Affiliations:** 1Department of Rehabilitation Medicine, Soonchunhyang University Cheonan Hospital, Soonchunhyang University College of Medicine, Cheonan 31151, Republic of Korea; 2Department of Rehabilitation Medicine, Gyeongsang National University Changwon Hospital, Gyeongsang National University College of Medicine, Changwon 51472, Republic of Korea; 3Department of Rehabilitation Medicine, Gyeongsang National University Hospital, Gyeongsang National University College of Medicine, Jinju 52727, Republic of Korea

**Keywords:** case report, cauda equina syndrome, electrical stimulation, electrodiagnosis, neurogenic bowel dysfunction

## Abstract

Neurogenic bowel dysfunction (NBD) is common in patients with cauda equina syndrome (CES). Previous studies have reported that electrical stimulation (ES) improves NBD but more neurophysiologic evidence is required. This case report describes a patient who experienced difficulty with defecation as a result of cauda equina syndrome (CES) that developed after a cesarean section performed 12 years ago under spinal anesthesia. The neurophysiological effects were assessed using the bulbocavernosus reflex (BCR) and electromyography (EMG). Two ES treatments, interferential current therapy and transcutaneous electrical stimulation, were used to stimulate the intestine and the external anal sphincter, respectively. The BCR results showed right-side delayed latency and no response on the left side. Needle EMG revealed abnormal spontaneous activities of the bilateral bulbocavernosus (BC) muscles. Electrodiagnostic testing revealed chronic bilateral sacral polyradiculopathy, compatible with CES. After treatment, the patient reported an improved perianal sensation, less strain and time for defecation than before, and satisfaction with her bowel condition. At the follow-up electrodiagnosis, the BCR latency was normal on the right side—needle EMG revealed reductions in the abnormal spontaneous activities of both BC muscles and re-innervation of the right BC muscle. Electrodiagnostic testing can offer insight into the neurophysiological effects of ES, which can help in understanding the mechanism of action and optimizing the therapy for patients with NBD.

## 1. Introduction

Neurogenic bowel dysfunction (NBD) is common in patients with spinal cord injury (SCI)—including cauda equina syndrome (CES), multiple sclerosis, spina bifida, Parkinson’s disease, and other neurological conditions [1]. Of those with CES, the prevalence of defecation symptoms has been reported as 38.1% [2]. NBD is caused by the loss of normal sensory or motor control of the gastrointestinal tract, which can lead to symptoms including constipation and fecal incontinence, negatively affecting the quality of life.

CES occurs most frequently following either a large lower lumbar disc herniation, prolapse, or sequestration. Other less common causes include spinal stenosis, epidural hematoma, infections, tumors, trauma, and prolapse after manipulation or chemonucleolysis [3]. Spinal anesthesia (SA) is a widely used technique for providing anesthesia and analgesia for various surgical procedures. However, in rare cases, it can also lead to neurological complications, including CES. The incidence of neurological complications following SA is estimated to be between 1 in 1000 and 1 in 100,000 procedures [4].

Management for NBD includes a bowel program, dietary modification, laxatives, electrical stimulation (ES), irrigation, and surgery [5]. ES is used to treat patients who do not respond to conservative therapy. It has been widely employed to treat neurological and musculoskeletal conditions, and urinary incontinence [6,7]. ES is gradually becoming employed to treat NBD. Of the various ES methods, interferential current therapy (IFC) features four surface electrodes delivering alternating medium-frequency electric current (1000–10,000 Hz) signals of slightly different frequencies [8]. Two currents that are slightly out of phase are delivered across the surface of the skin to produce an amplitude-modulated interference wave within tissues [8,9]. Transcutaneous electrical stimulation (TES) is the application of pulsed, direct-current, low-energy pulses (1–2 Hz) by a pair of electrodes to elicit muscle contractions [8]. The review by Southwell suggests that various electrical stimulation (ES) modalities have shown positive effects on relieving the symptoms of neurogenic bowel dysfunction (NBD), but there are limited high-quality published reports to support the effects of ES. The review also highlights the need for further understanding of the mechanisms of each modality to improve its effectiveness [10].

Herein, we report a case in which ES improved the clinical symptoms of a patient with NBD caused by CES which developed after SA. We assessed the neurophysiological effects using the bulbocavernosus reflex (BCR) and electromyography (EMG).

## 2. Case Description

A 40-year-old woman who had undergone a cesarean section under SA 12 years prior visited our Department of Rehabilitation Medicine complaining of defecation difficulty with hypoesthesia in the posterior part of her left lower extremity (LE) that persisted after presentation. She reported no bowel-filling sensation, and the need to apply strong abdominal pressure to defecate. Sometimes, symptoms such as uneasiness, headache, or perspiration developed. The average time required for defecation was 10~30 min and defecation was both irregular and less than once daily. Laxatives and other drugs had not worked and at times—less than once per week—she had to manually remove stool using her fingers. She also complained of flatus incontinence and dysuria but not urinary or fecal incontinence. Except for taking medication, she had never received any kind of rehabilitation. Neurological examination revealed the absence of the left ankle jerk reflex and perianal and buttock sensation, with atrophy and a reduced resting tone of the external anal sphincter (EAS) muscle. There was no pain or motor weakness of the lower back or LEs. Lumbar magnetic resonance imaging performed in another clinic revealed mild disc bulging at L4-5 but no spinal nerve root compression. She reported that she had no previous history of surgery, anesthesia, or nerve block in the spine, abdomen, pelvis, or lower extremities.

An electrodiagnostic examination (Viking Select; Nicolet, San Carlos, CA, USA) revealed a decreased motor response of the left tibial nerve. The bulbocavernosus reflex (BCR) results showed right-side delayed latency and no response on the left side. Needle EMG revealed abnormal spontaneous activities of the bilateral bulbocavernosus (BC) muscles and re-innervation of the left medial gastrocnemius and abductor hallucis muscles (Table 1). The diagnosis was chronic bilateral polyradiculopathy—at S1 and below on the left, and S2 and below on the right—compatible with CES. Anorectal manometry (ARM) revealed a decreased EAS, squeezing pressure, and reduced rectal compliance.

An IFC (Audiotron EF-150; OG GIKEN, Miyoshi, Okayama, Japan) was applied to the abdomen. The carrier frequency was 5 kHz, and the beat frequency was 0–360 Hz; the amplitude was adjustable (0–50 mA). As presented in previous studies, two electrodes, one from each channel, were bilaterally placed on the anterior abdominal wall below the costal margin. Two other electrodes (again, one from each channel) were placed crosswise on the patient’s back between T12 and L4. The current from each channel thus crossed within the abdomen and stimulated the bowel (Figure 1) [10,11]. The intensity was increased to the maximum without muscle contractions; the pain threshold was not attained and the treatment was thus tolerable. We also delivered TES (EST-1000; Stratek, Anyang, Gyeonggi-do, South Korea) to the EAS. The electrodes were placed on either side over the perianal area (Figure 2). TES was delivered at a low frequency (2 Hz) and the skeletal muscles twitched. The amplitude was adjusted to ensure muscle contraction without pain. She received 16 sessions at 30 min each (2 sessions/week).

At the end of treatment, the patient reported that perianal sensation had gradually improved during treatment. She felt a perianal pinching sensation which, although still difficult, required less strain during defecation than before. The average defecation time fell to about 10 min, and she could defecate daily and hold flatus. She was satisfied, although the symptoms had not been completely eliminated. At the 5-month follow-up, the BCR remained absent on the left, but BCR latency was normal on the right. Needle EMG revealed reductions in the abnormal spontaneous activities of both BC muscles and re-innervation of the right BC muscle (Table 1). She declined further examination or treatment because it was not easy for her to visit our hospital, but she reported that her bowel condition had not deteriorated.

## 3. Discussion

This is the first study to apply two ES treatments to a patient with NBD caused by CES after SA. It is also the first study to describe the neurophysiological effects of ES on NBD with reference to the BCR and EMG data.

SA features the injection of local anesthetics and other materials into the intrathecal space containing the cerebrospinal fluid surrounding the spinal cord [12]. It is generally safe and efficient and is commonly induced during cesarean section. However, uncommon side effects include hemodynamic changes, hypothermia, nausea and vomiting, back pain, headache, infection, total spinal anesthesia, and neurological injury [13]. CES typically develops when multiple lumbar and sacral nerve roots of the cauda equina are damaged. In cases after SA, CES is caused by direct needle trauma, direct exposure of the cauda equina roots of the neural canal to large doses of local anesthetic, the introduction of bacteria into the subarachnoid or epidural space, or (very rarely) epidural hematoma [12,13]. The rate of permanent neurological damage caused by CES is 0.3–1.2 per 100,000 SA inductions [4]. CES symptoms include pain in the back and/or LEs, LE weakness or paralysis, saddle anesthesia, and bladder, bowel, and sexual dysfunction.

NBD is a common problem—normal bowel function is lost because of nerve injury, neurological disease, or a congenital defect of the nervous system [14]. Fecal incontinence, chronic constipation, and fecal impaction are the main symptoms of NBD. Lower motor neuron bowel (LMNB) can be caused by polyneuropathy, conus medullaris, cauda equina lesions, pelvic surgery, or vaginal delivery (which can impair the bilateral pelvic nerve somatic innervation of the anal sphincter and reduce rectal tone and contraction) [15]. The clinical features of LMNB are chronic constipation and fecal impaction that is maximal in the rectum [5]. The colon is controlled by the central nervous system via the vagal and pelvic nerves, which carry sympathetic, parasympathetic, and somatic nerve fibers [16]. Sympathetic innervation inhibits colonic motility—parasympathetic innervation of the distal colon, rectum, and internal anal sphincter proceeds via the pelvic nerve through sacral segments S2~4 and increases motility [16]. The somatic nervous system allows voluntary regulation of the EAS and puborectalis muscles via the pudendal nerve through sacral segments S2~4 [16]. In the present case, NBD was caused by CES and was electrodiagnostically compatible with bilateral S2~4 polyradiculopathy. ARM revealed a decreased squeezing pressure of the EAS, low anal pressure during attempted defecation, and poor rectal sensation and compliance. The decreased rectal motility and sensation and the faulty anal sphincter mechanism induced difficult bowel evacuation, a long defecation time, and fecal impaction. Atrophy and a reduced resting tone of the EAS muscle were also observed, together with a reduced BCR response on EMG. The findings seem to be attributable to damage to the S2~4 segments, which created deficits in parasympathetic and somatic innervation.

ES treatments have shown promise in treating bowel dysfunction, but the evidence is often limited and the mechanisms of action are not fully understood. Clinical studies have reported positive therapeutic effects of ES treatments [10], and experimental studies have attempted to explain the underlying mechanisms. For example, electroacupuncture at Zusanli (ST36) has been shown to downregulate colonic neuronal nitric oxide synthase (nNOS) in rats with spinal cord injury (SCI), potentially improving bowel dysfunction. [17]. Sacral nerve stimulation (SNS) has been found to improve defecation reflex and promote the recovery of intestinal transit function by reducing nNOS expression in the colon and sacral cord of SCI rats [18]. SNS has also been shown to increase serotonin receptors in the sacral defecation center and increase serotonin content in the colon, leading to improved defecation reflex and colonic transmission function in rats with SCI [19]. Direct electrical colonic stimulation has been found to activate the colon along its neuromuscular structure, thus reducing colonic transit time. [20] In Yik et al.’s study, they found that longer stimulation duration (2 weeks) of IFC improved whole gut transit in pig models, but studying the mechanism of action in animals is challenging due to the cost and difficulty of handling animals [10]. The mechanism of action of TES is also unclear [21], including its electrophysiological mechanism [22].

Unlike other ES treatments including SNS, electroacupuncture, and direct intestinal stimulation, IFC and TES are noninvasive techniques applied to the skin. Transcutaneous IFC stimulation of the abdominal area probably stimulates local skin nerve fibers; deeper stimulation may activate sympathetic and parasympathetic outflows to the intestine and nerves within the intestine [9]. However, such treatment is unlikely to directly stimulate the pelvic floor or EAS [9]. TES—using electrodes placed over the perineum—directly stimulates the pelvic floor and causes sub-skin skeletal muscle fibers to repeatedly contract and relax between pulses [8,11]. In this study, both types of ES were used to stimulate the intestine and the EAS, respectively, while earlier studies targeted single bodily regions [9,10,11]. The principle of electrical stimulation is that it can activate sensory and motor nerves in the skin, spinal nerves, sympathetic and parasympathetic nerves, enteric nerves in the bowel wall or pacemaker cells in the intestine, and intestinal muscle cells [8]. In the present study, IFC therapy activated the autonomic nervous system and nerves within the intestine. Direct TES of the EAS increased the pressure afforded by that muscle and sensory input to the perineum via the pudendal nerve. The improved sacral reflex seemed to influence neural bowel control and thus contribute to clinical improvement.

The BCR is a polysynaptic reflex mediated via the S2~4 spinal cord segments and is useful when exploring the integrity of the sacral reflex arc [23,24]. Assessments of the BCR and the EAS muscle are required for patients with suspected S2~4 nerve root lesions complaining of bladder, bowel, and/or sexual dysfunction [25]. Although a physical examination yields clinically important information, some features may be unclear, and the results will depend on the examiner. Electrodiagnostic tests objectively assess lesional neurophysiology. The BCR is electrically examined by stimulating the dorsal nerve of the penis or clitoris and recording the response from the BC muscle [26]. This reflex, and pelvic floor and sphincter muscle EMG data, are the most useful sacral neurophysiological tests [25]. The BCR demonstrates the continuity of the S2~4 reflex arc better than physical examination does [27]. We used EMG to examine the BC muscle but not the EAS muscle. We sought to minimize the discomfort that would be caused by the insertion of an additional needle and the time required to examine a sensitive part of the body. The EAS muscle is principally responsible for voluntary defecation control but also forms a single functional unit with other pelvic floor muscles, including the BC muscle with which it shares contractile activity [28,29]. Thus, the BCR combined with BC muscle EMG data was considered adequate when evaluating this neurological condition of our patient. The electrodiagnostic findings, including recovery of the BCR response on one side, reduced denervation, and re-innervation of the BC muscle, indicate improved activation of the pelvic floor muscles and the sacral reflex arc after ES treatments. These seem to have improved bowel function and symptoms.

This case report had certain limitations. Changes in gastrointestinal function and possible associations of such changes with neurophysiological changes could not be confirmed because follow-up ARM was not performed. In addition, as the follow-up was short, we do not know whether the therapeutic effect of ES persisted. However, the therapeutic utility of ES was identified via neurophysiological changes evident in electrodiagnosis. A study with a larger sample size, longer follow-up, and that features more assessments is required.

## 4. Conclusions

In the case of a patient with NBD and CES after SA, electrodiagnosis can provide insight into the neurophysiological effects of ES, which can aid in understanding the mechanism of action and optimizing the therapy for patients with NBD. Additionally, it can help in the diagnosis and follow-up of the conditions. Accumulating more evidence of the neurophysiological effect of ES through electrodiagnostic testing could help in better understanding the mechanism and effectiveness of ES for patients with NBD. This could lead to the development of more targeted and effective ES protocols for NBD and may also lead to its wider application in clinical practice. However, further studies are needed to confirm the effectiveness and safety of ES for NBD and to identify optimal stimulation parameters and patient selection criteria.

## Figures and Tables

**Figure 1 medicina-59-00588-f001:**
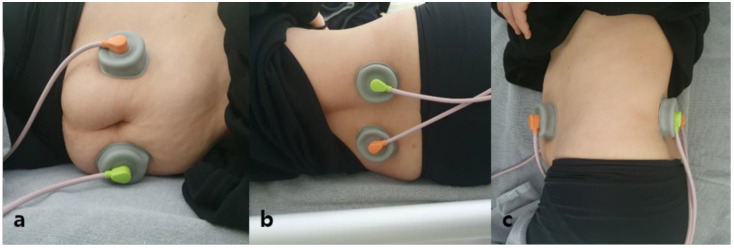
Position of four electrodes from two-channel interferential current therapy stimulation: (**a**) anterior view; (**b**) posterior view and (**c**) lateral view.

**Figure 2 medicina-59-00588-f002:**
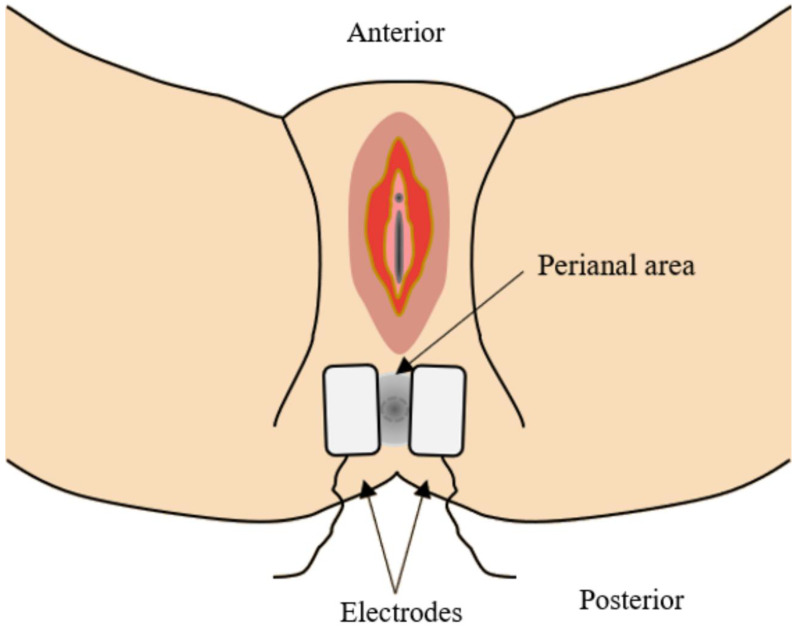
Diagram of a single pair for electrical stimulation of external anal sphincter, as used in this study.

**Table 1 medicina-59-00588-t001:** Results of Nerve Conduction study, Electromyography, and Bulbocaverouns Reflex.

**Motor Nerve Conduction Study**						
Nerve/Site	Onset Latency (ms)	Peak Amplitude (mV)		Conduction Velocity (m/s)		
Tibial (Rt./Lt.)						
Ankle/AH	3.59/3.02	18.8/4.8				
Knee/AH	9.48/8.85	16.9/4.2		58/58		
**Sensory Nerve Conduction Study**						
Nerve/Sites	Onset Latency (ms)	Peak Latency (ms)		Peak Amplitude (mV)		
Sural/Ankle (Rt./Lt.)	1.51/1.67	2.34/2.24		15.4/11.2		
**BCR (Rt./Lt.)**						
Onset Latency (ms)	Initial	75.1/NR		Follow-up (5 mo)		32.9/NR
**Needle Electromyography**						
Muscle	Abnormal Spontaneous Activities		MUAP			
	Fib	PSW	Poly	Amp	Dur	Interfer.
Left AH	(−)	(−)	(−)	Inc	N	Reduced
Left GCM	(−)	(−)	Inc	(−)	N	N
Left BC (Initial)	2+	(−)	(−)	N	N	N
Left BC (Follow-up)	1+	(−)	(−)	N	N	N
Right BC (Initial)	2+	(−)	(−)	N	N	N
Right BC (Follow-up)	(−)	(−)	(−)	Inc	N	N

AH abductor hallucis, BCR bulbocarvenous reflex, NR no response, GCM medial gastrocnemius, BC bulbocarvenous, MUAP motor unit action potential, Fib fibrillation, PSW positive sharp wave, Poly polyphasia, Amp amplitude, Dur duration, Interfer. interference pattern, N normal.

## Data Availability

The data presented in this study are available on request from the corresponding author.

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
