# Peer review of "Neurophysiological Effects of Electrical Stimulation on a Patient with Neurogenic Bowel Dysfunction and Cauda Equina Syndrome after Spinal Anesthesia: A Case Report"

_medicina, 2023, doi:10.3390/medicina59030588_

Round 1
Reviewer 1 Report
This case report applied two electrical stimulation treatments, interferential current therapy and transcutaneous electrical stimulation treatments to a patient with NBD caused by CES after SA, and showed a neurophysiological effect. I am just sorry that there is only one case and the follow-up test was not performed. Anyway, this paper provided the new evidence and might be valuable for patients with NBD. Is there any animal model experiments that can support this therapeutic effect?
Author Response
RESPONSE
I am glad to hear the reviewer’s feedback. Your first point is the limitation of our research. We agree that Further studies are needed to confirm the effectiveness and safety of ES for NBD.
According to the reviwer’s comment, we have added more explanation about the experimental result in animal models to explain the action mechanisms of ES. . (Page 6, Para 3 ~ Page 7, Para 1, Line 156 -171):
“ES treatments has shown promise in treating bowel dysfunction, but the evidence is often limited and the mechanisms of action are not fully understood. ~ The mechanism of action of TES is also unclear [8], including its electrophysiological mechanism [9]”

Reviewer 2 Report
This is a research with certain depth and value. The study provided a prophetic clinical finding that identifiedSpinal anesthesia (SA) is safe and efficient and is commonly induced during cesarean section. The rare neural complications include cauda equina syndrome (CES) attributable to direct needle trauma, direct exposure of the cauda equina roots of the neural canal to large doses of local anesthetic, introduction of bacteria into the subarachnoid or epidural space, or (very rarely) epidural hematoma. Neurogenic bowel dysfunction (NBD) is common in patients with CES. The symptoms include constipation and fecal incontinence that negatively affect quality-of-life. Management includes a bowel program, dietary modification, laxatives, electrical stimulation (ES), irrigation, and surgery . ES is used to treat patients who do not respond to conservative therapy. It has been widely employed to treat neurological and musculoskeletal conditions, and urinary incontinence and is gradually becoming employed to treat NBD. Of the various ES methods, interferential current (IFC) therapy features four surface electrodes delivering alternating medium-frequency electric current (1000–10,000 Hz) signals of slightly different frequencies. Two currents that are slightly out of phase are delivered across the surface of the skin to produce an amplitude-modulated interference wave within tissues . Transcutaneous electrical stimulation (TES) is the application of pulsed, direct-current low-energy pulses (1–2 Hz) by a pair of electrodes to elicit muscle contractions. Some studies have found that ES improves NBD but more evidence is required .
1. the results or data can support any conclusions shown directly or otherwise publicly available according to the standards of the field
2. the conclusions are a reasonable extension of the results.
Other comments:
1. Whether the clinical sample can be further expanded?
2. Whether the patient's other medical conditions were taken into account when screening the sample?
3. Pay attention to the beauty of color matching in some pictures.
4. You can pay due attention to the quality of the picture.
5. Is it better to supplement the conclusions of some basic experiments to verify?
6. Can the experimental data be further processed to have better results?
Author Response
1. the results or data can support any conclusions shown directly or otherwise publicly available according to the standards of the field
RESPONSE 1)
I am glad to hear the reviewer’s feedback.
According to the reviwer’s comment, we have added more explanation about the existing accepted principle of the mechanism of ES (Page 7, Line 180-183) and clarified the conclusion (Page 8~9, Line 216-225):
“The principle of electrical stimulation is that it can activate sensory and motor nerves in the skin and spinal nerves, sympathetic and parasympathetic nerves, enteric nerves in the bowel wall or pacemaker cells in the intestine, and intestinal muscle cells [8]. In present study,”
“In the case of a patient with NBD and CES after SA, electrodiagnosis can provide insight into the neurophysiological effects of ES, which can aid in understanding the mechanism of action and optimizing the therapy for patients with NBD. Additionally, it can help in the diagnosis and follow-up of the conditions. Accumulating more evidence for the neurophysiological effect of ES through electrodiagnostic testing could help in better understanding the mechanism and effectiveness of ES for patients with NBD. This could lead to the development of more targeted and effective ES protocols for NBD, and may also lead to its wider application in clinical practice. However, further studies are needed to confirm the effectiveness and safety of ES for NBD and to identify optimal stimulation parameters and patient selection criteria.”
2. the conclusions are a reasonable extension of the results.
RESPONSE 2)
As mentioned earlier, we have modified the conclusions more reasonable to avoid leaps (Page 8~9, Line 216-225):
Other comments:
1. Whether the clinical sample can be further expanded?
RESPONSE 3)
Yes. Although we have identified the therapeutic utility of ES via neurophysiological changes evident in electrodiagnosis, there is still a lack of evidence to understand the mechanism and effectiveness of ES for patients with NBD. If the circumstances allow, we have a plan to recruit more NBD patients regardless of its cause and conduct electrodiagnosis before and after ES treatments.
2. Whether the patient's other medical conditions were taken into account when screening the sample?
RESPONSE 4)
The electrodiagnostic testing shown in this study is to examine perineum area. TES is also applied at that area. Thus, medical conditions including skin problem, fecal or urinary incontinence and use of urinary catherization should be considered when screening patients. In addition, in SCI patients who have decreased or no sense at perineum, care should be taken during examination or treatment not to avoid unexpected secondary injury or problem.
3. Pay attention to the beauty of color matching in some pictures.
RESPONSE 5)
We have modified the color matching of Figure S2.
4. You can pay due attention to the quality of the picture.
RESPONSE 6)
We have modified the resolution of figures.
5. Is it better to supplement the conclusions of some basic experiments to verify?
RESPONSE 7)
According to the reviwer’s comment, we have added more explanation about the experimental result in animal models to explain the action mechanisms of ES. (Page 6, Para 3 ~ Page 7, Para 1, Line 156 -171):
“ES treatments has shown promise in treating bowel dysfunction, but the evidence is often limited and the mechanisms of action are not fully understood. ~ The mechanism of action of TES is also unclear [8], including its electrophysiological mechanism [9]”
6. Can the experimental data be further processed to have better results?
RESPONSE 8)
According to the opinion of Southwell et al., who conducted a number of studies about electrical stimulation treatments including animal model, studying the mechanism of action in animals is challenging due to cost and difficulty of handling animals. However, if the circumstances allow, we think that further animal studies to understand the action mechanisms and effectiveness of transcutaneous electrical stimulation should be tried.

Reviewer 3 Report
EAS in the abstract should be explained. It is presented as acronym
Author Response
RESPONSE
I am glad to hear the reviewer’s feedback. We have corrected the word as follows:
"EAS-> External anal sphincter"

Reviewer 4 Report
Thank you for the opportunity to review this important manuscript.
This is an excellent description of a case and therapeutic success.
My only question, which I find very important, is how authors associate cauda equina syndrome with spinal anesthesia 12 years ago. This is very important because then there is no association and this case should be, completely rewritten in the initial part and the title, of course.
Author Response
RESPONSE
I am glad to hear the reviewer’s feedback. She reported that defecation difficulty with hypoesthesia in the posterior part of her left lower extremity (LE) that persisted after presentation to our department visit. She also reported that she had no previous history of surgery or anesthesia in the spine, abdomen, pelvis, and lower extremities. The electodiagnosis findings indicated chronic state of polyradiculopathy. Additionally, lumbar magnetic resonance imaging revealed mild disc bulging at L4-5 but no spinal nerve root compression, which was not associated with her symptoms and EMG findings. Based on her medical history and examination findings, we judged that cauda equina syndrome had occurred after cesarean section under under spinal anesthesia. We have added more explanation abouth her medical history (Page 4, Para 2, Line 86 -87):
"She reported that she had no previous history of surgery or anesthesia in the spine, abdomen, pelvis, and lower extremities after presentation."

Round 2
Reviewer 4 Report
Again I do not know when was the spinal anesthesia applied. If that was 12 years ago then there is no connection. Also, please put that finding in the abstract
Author Response
RESPONSE
I apologize for my inadequate response.
Her defecation symptoms had newly begun after presentation. According to her medical history and examination findings, there were no other possible causes to provoke cauda equina syndrome, and spinal anesthesia during a cesarean section performed 12 years ago seems to have contributed to the development of cauda equina syndrome. Based on the results of electromyography (EMG) and MRI, CES also seems to have occurred 12 years ago and was related to spinal anesthesia at that time.
We have revised the sentence to make the timeline clearer. (Page 2, Line 23 -25):
“This case report describes a patient who experienced difficulty with defecation as a result of cauda equina syndrome (CES) that developed after a cesarean section performed 12 years ago under spinal anesthesia.”